Mammals from ‘down under’: a multi-gene species-level phylogeny of marsupial mammals (Mammalia, Metatheria)

May-Collado Laura J. lmaycollado@gmail.com
Kilpatrick C. William
Agnarsson Ingi
Department of Biology, University of Vermont , Burlington, VT , USA
Huang Xiaolei
Electronic publication date: 2015 Feb 26
Publication date: 2015
Volume: 3
Electronic Location ID: e805
Received 2014 Nov 24; Accepted 2015 Feb 9
Copyright: © 2015 May-Collado et al.
Copyright year: 2015
Copyright holder: May-Collado et al.
License: This is an open access article distributed under the terms of the Creative Commons Attribution License, which permits unrestricted use, distribution, reproduction and adaptation in any medium and for any purpose provided that it is properly attributed. For attribution, the original author(s), title, publication source (PeerJ) and either DOI or URL of the article must be cited.
License URL: https://creativecommons.org/licenses/by/4.0/

Keywords: Australidelphia, Ameridelphia, Microbiotheria, Dasyuromorphia, Didelphimorphia, Diprotodontia, Peramelemorphia, Notoryctemorphia, Paucituberculata

Funding: University of Vermont National Science Foundation DEB 1314749 Funding for this work came from the University of Vermont and National Science Foundation DEB grant 1314749 to Ingi Agnarsson. The funders had no role in study design, data collection and analysis, decision to publish, or preparation of the manuscript.

==============================
Marsupials or metatherians are a group of mammals that are distinct in giving birth to young at early stages of development and in having a prolonged investment in lactation. The group consists of nearly 350 extant species, including kangaroos, koala, possums, and their relatives. Marsupials are an old lineage thought to have diverged from early therian mammals some 160 million years ago in the Jurassic, and have a remarkable evolutionary and biogeographical history, with extant species restricted to the Americas, mostly South America, and to Australasia. Although the group has been the subject of decades of phylogenetic research, the marsupial tree of life remains controversial, with most studies focusing on only a fraction of the species diversity within the infraclass. Here we present the first Methaterian species-level phylogeny to include 80% of the extant marsupial species and five nuclear and five mitochondrial markers obtained from Genbank and a recently published retroposon matrix. Our primary goal is to provide a summary phylogeny that will serve as a tool for comparative research. We evaluate the extent to which the phylogeny recovers current phylogenetic knowledge based on the recovery of “benchmark clades” from prior studies—unambiguously supported key clades and undisputed traditional taxonomic groups. The Bayesian phylogenetic analyses recovered nearly all benchmark clades but failed to find support for the suborder Phalagiformes. The most significant difference with previous published topologies is the support for Australidelphia as a group containing Microbiotheriidae, nested within American marsupials. However, a likelihood ratio test shows that alternative topologies with monophyletic Australidelphia and Ameridelphia are not significantly different than the preferred tree. Although further data are needed to solidify understanding of Methateria phylogeny, the new phylogenetic hypothesis provided here offers a well resolved and detailed tool for comparative analyses, covering the majority of the known species richness of the group.

Introduction

The infraclass Metatheria contains seven mammalian orders that share a reproductive strategy, giving birth to undeveloped young and having prolonged investment in lactation (Aplin & Archer, 1987). The group includes the familiar Australian megafauna, such as kangaroos and koalas, as well as some enigmatic mammals such as wombats, the Tasmanian devil, and the unique South American Monito del Monte. Many species are at risk of extinction (Isaac et al., 2007), and two marsupial families have recently gone extinct: Thylacinidae (1936) and Chaeropodidae (∼1950). Marsupials have a rather unusual geographic distribution, mostly inhabiting Australasia and South America (Nilsson et al., 2004), with a few genera having relatively recently crossed the Panamanian isthmus and one species (the Virginia opossum) reaching northern North America. Most prior phylogenetic work has suggested that marsupials colonized Australia twice via Antarctica/South America during the breakup of Gondwanaland (Nilsson et al., 2004). However, a recent study supports the monophyly of the Australasian marsupials, and thus that marsupials reached Australasia in a single migration event (Nilsson et al., 2010) and then diversified with over 200 extant species in the region.

Marsupials are typically classified into two major cohorts, the Australidelphia and the Ameridelphia (Aplin & Archer, 1987; Marshall, Case & Woodburne, 1990), based in part on differences of the ankle joints (Szalay, 1982). Australidelphia consists of five orders: Dasyuromorphia (carnivorous marsupials and marsupial mice), Peramelemorphia (bilbies and bandicoots), Notoryctemorphia (marsupial moles), Diprotodontia (koalas, wombats, kangaroos, and possums), and the South American order Microbiotheria (monito del monte). Ameridelphia consist of two orders: Didelphimorphia (opossums) and Paucituberculata (shrew oposums), mainly distributed in South America (Gardner, 2005a; Gardner, 2005b). Most recent studies, however, have shown that Ameridelphia is non-monophyletic and thus this classification is inconsistent with phylogenetic knowledge (Horovitz & Sánchez-Villagra, 2003; Nilsson et al., 2010).

While the phylogenetics of marsupials has received much attention in recent years, the marsupial tree of life remains incompletely resolved (Nilsson et al., 2010). Most studies have focused on solving phylogenetic relationships within orders (Krajewski et al., 1997; Blacket et al., 1999; Jansa, Forsman & Voss, 2006; Meredith, Westerman & Springer, 2008a; Meredith, Westerman & Springer, 2008b; Frankham, Handasyde & Eldridge, 2012), while the root of the marsupial tree and the relationships among the four Australasian and three South American marsupial orders have not been resolved conclusively with standard sequence data or morphological evidence (Springer et al., 1998; Horovitz & Sánchez-Villagra, 2003; Nilsson et al., 2003; Asher, Horovitz & Sánchez-Villagra, 2004; Nilsson et al., 2010). Particularly contentious has been the early branching pattern within Metatheria. For example, it is unclear whether Paucituberculata or Didelphimorphia are the sister group to the remaining marsupials (Meredith, Westerman & Springer, 2009). Furthermore, the phylogenetic position of the enigmatic Microbiotheria, represented by only one American species ‘monito del monte’ (Dromiciops gliroides), differs among studies (Springer et al., 1998; Burk et al., 1999; Amrine-Madsen et al., 2003; Nilsson et al., 2003; Nilsson et al., 2004), but is usually placed among the Australasian marsupials, implying a biogeographical history that is not straightforward to interpret. However, Nilsson et al. (2010) provided an analysis of retroposon insertions that provide a powerful alternative to sequence data, especially to resolve deeper level relationships. They find support for an intuitively pleasing hypothesis placing all Australasian marsupials in a single clade, as a sister group to Microbiotheria. They also provide strong evidence that Didelphimorphia forms the sister group of the remaining marsupials. A few other studies have studied marsupial species-level relationships mainly within small taxonomic groups (families and subfamilies, genera) or employing relatively sparse taxon sampling (Krajewski et al., 1997; Krajewski et al., 2012; Blacket et al., 1999; Jansa, Forsman & Voss, 2006; Meredith, Westerman & Springer, 2008a; Meredith, Westerman & Springer, 2008b; Malekian et al., 2010; Frankham, Handasyde & Eldridge, 2012).

Detailed species-level phylogenies underlie modern comparative studies (Harvey & Pagel, 1991). In general, the statistical power of comparative methods increases as taxon sampling and resolution improves. In addition, many methods in the toolkit of comparative biology perform best when branch length estimates are available (Felsenstein, 2004; Bollback, 2006). At present the most detailed species-level phylogeny of marsupials available is based on a supertree including approximately 260 taxa (Cardillo et al., 2004). This phylogeny has already proven to be a high utility tool, underlying various comparative analyses (Weisbecker et al., 2008; Sánchez-Villagra et al., 2008; Flores, Abdala & Giannini, 2013). Yet, supertrees are essentially summary hypotheses that are stitched together based on smaller phylogenetic studies, and where these are lacking is in taxonomy. Thus, supertrees are constrained by the available input data, in part a summary of opinion rather than primary phylogentic data, often lacking full resolution and typically without accurate estimates of branch lengths. Here we present a species-level phylogeny with branch-length information, including 276 marsupial species, with the primary goal of providing an additional tool for taxonomy, phylogenetic estimation of conservation priorities, and comparative hypothesis testing. We evaluate the ‘reliability’ of the phylogeny based on the recovery of numerous benchmark clades-previously supported clades and undisputed taxonomic groups (Agnarsson & May-Collado, 2008).

Material and Methods

Data and phylogenetic analyses

Sequences for five mitochondrial genes cytochrome b (cytb), ribosomal RNAs 12S and 16S, cytochrome oxidase I (COI), Nicotinamide adenine dinucleotide dehydrogenase subunit 2 (NADH2) and six nuclear genes apolipoprotein B (ApoB), von Willebrand factor (vWF), interphotoreceptor retinoid-binding protein (IRBP), recombination activating gene 1 (Rag1), Breast cancer 1(BRCA1), and Protamine 1 (PRM1) were downloaded from GenBank for 271 extant and five extinct species (Table S1). Sequences were downloaded via Mesquite 2.75v (Maddison & Maddison, 2011), aligned in Mafft (Katoh et al., 2002) online (http://www.ebi.ac.uk/) and then reintroduced to Mesquite and manually inspected. When different genes were available for different subspecies we created chimeras to represent the species. We selected 19 outgroups species representing the diversity of Mammalia, including the orders Monotremata, Pilosa, Pholidota, Chiroptera, Rodentia, Dermoptera, Carnivora, Erinaceomorpha, Soricomorpha, Scandentia, Perissodactyla, the supraorder Cetartiodactyla and the magnaorder Afrotheria. We created several data partitions for sensitivity analyses to explore potential data conflict and source of support, or the lack of support for the phylogeny. Conflict and lack of support might be expected because (1) mitochondrial and nuclear genes often differ in estimation of deeper level clades, (2) we identified some alignment issues in the protamine data suggesting potential problems with the source sequences, and (3) the data from GenBank are fragmentary, in that most species are missing part of the character data, with some species having less than 10% data coverage. These partitions consisted of the following matrices: full concatenated matrix (298 taxa, 18,723 bp), full concatenated excluding species with less than 10% data coverage (251 taxa, 18,723 bp) hereafter referred to as ‘focal’analysis, full concatenated minus protamine (296 taxa, 17,663 bp), concatenated mtDNA (282 taxa, 6,857 bp), concatenated nuDNA (242 taxa, 11,866 bp), concatenated nuDNA excluding protamine (237 taxa, 10,806 bp), and the concatenated the 251 taxa full partitions with a retroposon matrix of Nilsson et al. (2010). Finally, to test if results are sensitive to alignment ambiguities, we created a matrix excluding non-protein coding genes (12S and 16S) plus the ambiguously aligned protamine, for a total of eigth partitions.

Table 1 Summary of benchmark clades supported by multiple studies.

The list of references accompanying each clade is meant to be representative, not an exhaustive review of supporting studies.

Benchmark clade	Description	Morphology	Molecular	
Australidephia	This superorder contains all Australian marsupials and a single species from South America (monito del monte, Dromiciops gliroides)	Horovitz & Sánchez-Villagra (2003), Asher, Horovitz & Sánchez-Villagra (2004), Ladèveze & de Muizon (2010)	Kirsch et al. (1991), Colgan (1999), Palma & Spotorno (1999), Amrine-Madsen et al. (2003), Baker et al. (2004), Nilsson et al. (2004), Phillips et al. (2006), Beck (2008), Meredith, Westerman & Springer (2008a), Meredith, Westerman & Springer (2008b), Meredith, Westerman & Springer (2009), Nilsson et al. (2010), Westerman, Meredith & Springer (2010)	
Diprotodontia	This is the largest order of marsupials and is distinguished from other marsupials by having syndactylous digits and two procumbent lower incisors (diprotodont)	Horovitz & Sánchez-Villagra (2003), Asher, Horovitz & Sánchez-Villagra (2004)	Baverstock, Kri & Birrell (1990), Burk et al. (1999), Colgan (1999), Osborne, Christidis & Norman (2002), Amrine-Madsen et al. (2003), Phillips et al. (2006), Meredith, Westerman & Springer (2008a), Meredith, Westerman & Springer (2008b), Meredith, Westerman & Springer (2009), Munemasa et al. (2008), Phillips & Pratt (2008), Nilsson et al. (2010), Westerman, Meredith & Springer (2010)	
Phalangeriformes	This suborder of Diprotodontia contains medium sized arboreal marsupials from Australia, New Guinea and Sulawesi	Flannery (1987), Springer & Woodburke (1989)	Springer & Kirsch (1989), Springer & Kirsch (1991), Amrine-Madsen et al. (2003), Phillips & Pratt (2008), Meredith, Westerman & Springer (2009)	
Phalangeroidea	This superfamily of Phalangeriformes contains two families Phalangeridae and Burramyidae		Colgan (1999), Meredith, Westerman & Springer (2008a), Meredith, Westerman & Springer (2008b), Westerman, Meredith & Springer (2010)	
Phalangeridae	This family of Phalangeroidea contains brushtail possums and cuscuses	Hughes (1965)	Osborne, Christidis & Norman (2002), Baker et al. (2004), Kavanagh et al. (2004), Raterman et al. (2006), Beck (2008), Meredith, Westerman & Springer (2009).	
Burramyidae	This family of Phalangeroidea contains pygmy possums	Archer (1984)	Baverstock, Kri & Birrell (1990), Edwards & Westerman (1995), Osborne, Christidis & Norman (2002), Beck (2008)	
Petauroidea	This superfamily of Phalangeriformes contains four families: Pseudocheiridae, Petauridae, Tarsipedidae, and Acrobatidae		Osborne, Christidis & Norman (2002), Amrine-Madsen et al. (2003), Kavanagh et al. (2004), Meredith, Westerman & Springer (2008a), Meredith, Westerman & Springer (2008b), Phillips & Pratt (2008), Meredith et al. (2010), Westerman, Meredith & Springer (2010).	
Pseudocheiridae	This family of the superfamily Petauroidea contains ringtail possums	Archer (1984), Springer (1993)	Hayman & Martin (1974), Baverstock, Kri & Birrell (1990), Westerman, Janczewski & O’Brien (1990), Baverstock et al. (1990), Osborne & Christidis (2001), Meredith et al. (2010)	
Petauridae	This family of the superfamily Petauroidea contains gliders, Leadbeater’s possum, and the striped possum and trioks	Aplin & Archer (1987), Smith (1984)	Kirsch & Calaby (1977), Mckay (1984), Baverstock, Kri & Birrell (1990), Osborne & Christidis (2001), Meredith et al. (2010).	
Acrobatidae	This family of the superfamily Petauroidea contains feather-tailed gliders and feather-tailed possum		Baverstock, Kri & Birrell (1990), Baker et al. (2004).	
Macropodiformes	This suborder of Diprotodontia contains kangaroos, wallabies, and allies (bettongs, potaroos, and rat kangaroos)	Ride (1961), Case (1984), Flannery (1987)	Kirsch (1977), Burk & Springer (2000), Kavanagh et al. (2004), Meredith, Westerman & Springer (2008a), Meredith, Westerman & Springer (2008b), Meredith, Westerman & Springer (2009), Westerman, Meredith & Springer (2010)	
Macropodoidea	This superfamily of Macropodiformes consists of two families the Macropodidae and Potoroidae that form a clade distinct from the rat kangaroo, family Hypsiprymnodontidae		Baverstock, Kri & Birrell (1990), Burk, Westerman & Springer (1998), Colgan (1999), Osborne, Christidis & Norman (2002), Meredith, Westerman & Springer (2008b), Phillips & Pratt (2008).	
Macropodidae	This family of Macropodoidea contains the major diversity of marsupial herbivores, including kangaroos, wallabies, tree-kangaroos and several others	Horovitz & Sánchez-Villagra (2003), Prideaux & Warburton (2010)	Baverstock, Kri & Birrell (1990), Burk, Westerman & Springer (1998), Baker et al. (2004), Kavanagh et al. (2004), Meredith, Westerman & Springer (2008b), Meredith, Westerman & Springer (2009).	
Potoridae	This family of Macropodoidea contains bettongs	Archer (1984), Flannery, Archer & Plane (1984), Flannery (1989)	Baverstock, Kri & Birrell (1990), Sanclair & Westerman (1997), Burk, Westerman & Springer (1998), Meredith, Westerman & Springer (2008b).	
Vombatiformes	This suborder of Diprotodontia consists of two families: Phascolarctidae and Vombatidae	Hughes (1965)	Baverstock, Kri & Birrell (1990), Burk et al. (1999) (mtDNA), Osborne, Christidis & Norman (2002), Amrine-Madsen et al. (2003), Baker et al. (2004) (RAG1), Kavanagh et al. (2004), Meredith, Westerman & Springer (2008a), Meredith, Westerman & Springer (2008b), Meredith, Westerman & Springer (2009), Phillips & Pratt (2008), Westerman, Meredith & Springer (2010).	
Dasyuromorphia	This order of marsupials contains most of Australian carnivorous marsupials consisting of three families: Dasyuridae, Myrmecobiidae, and Thylacinidae	Wroe et al. (2000), Horovitz & Sánchez-Villagra (2003), Ladèveze & de Muizon (2010)	Burk et al. (1999), Amrine-Madsen et al. (2003), Kavanagh et al. (2004), Beck (2008), Meredith, Westerman & Springer (2009), Nilsson et al. (2010), Westerman, Meredith & Springer (2010).	
Dasyuridae	This family of Dasyuromorphia consists of terrestrial and arboreal species, many of which lack a pouch	Wroe et al. (2000), Asher & Kirsch (2006)	Westerman & Woolley (1990), Colgan (1999), Baker et al. (2004), Meredith, Westerman & Springer (2008a), Meredith, Westerman & Springer (2008b).	
Notoryctemorphia	This order of marsupials contains two species of marsupial moles, Notoryctescaurinus and N typhlops	Aplin & Archer (1987), Archer et al. (2011)	Baverstock, Kri & Birrell (1990), Springer et al. (1998), Nilsson et al. (2010)	
Peramelemorphia	This order of marsupials consists of three families: Peramelidae, Chaeropodidae and Thylacomidae	Wroe et al. (2000), Horovitz & Sánchez-Villagra (2003), Asher, Horovitz & Sánchez-Villagra (2004), Ladèveze & de Muizon (2010)	Burk et al. (1999), Amrine-Madsen et al. (2003), (Asher, Horovitz & Sánchez-Villagra, 2004), Baker et al. (2004), Kavanagh et al. (2004), Beck (2008), Meredith, Westerman & Springer (2008a), Meredith, Westerman & Springer (2008b), Meredith, Westerman & Springer (2009), Nilsson et al. (2010), Westerman, Meredith & Springer (2010), Westerman et al. (2012).	
Peramelidae	This family of the Peramelemorphia contains bandicoots and echymiperas		Phillips et al. (2006), Meredith, Westerman & Springer (2008a), Meredith, Westerman & Springer (2008b), Westerman et al. (2012).	
Paucituberculata	This order of shrew opossums is represented by a single family Caenolestidae	Marshall (1980), Sánchez-Villagra (2001), Asher & Kirsch (2006), Ladèveze & de Muizon (2010), Abello (2013)	Amrine-Madsen et al. (2003), Nilsson et al. (2004), Phillips et al. (2006), Meredith, Westerman & Springer (2008a), Meredith, Westerman & Springer (2008b), Nilsson et al. (2010)	
Didelphimorphia	This order of new world marsupials diversified mainly in South America and consists of a single family Didelphidae	Horovitz & Sánchez-Villagra (2003), Asher & Kirsch (2006), Ladèveze & de Muizon (2010)	Burk et al. (1999), Amrine-Madsen et al. (2003), Baker et al. (2004), (Kavanagh et al., 2004), Nilsson et al. (2004), Phillips et al. (2006), Meredith, Westerman & Springer (2008a), Meredith, Westerman & Springer (2008b), Nilsson et al. (2010)	

Table 2 Results from the Shimodaira-Hasagawa evaluating alternative topologies compared to the focal analysis (concatenated analysis removing taxa with <10% character data cover).

Tree constraint	-lnL	Diff-lnL	P	
Ameridelphia monophyletic	435351.2	5.84062	0.463	
Didelphimorphia is sister to the remaining Marsupialia	435353.4	2.22915	0.725	
Australian Australidelphia monophyletic (that is Microbiotheriidae is sister to other Australidelphia) (Nilsson et al., 2010)	435361.1	9.91611	0.263	

The appropriate models for the Bayesian analysis were selected with jModeltest (Darriba et al., 2012) using the AIC criterion (Posada & Buckley, 2004) with a UPGMA tree chosen as the basis for Modeltest. The selected models of sequence evolution for Cyt b, 12S, 16S, ApoB, IRBP, RAG 1, NADH2, and BRCA 1 was GTR + I + G, and for COI and Protamine 1 the selected models were HKY + I + G and HKY + G, respectively. The retroposon partition was analyzed under a parsimony model, and alternatively using GTR + I + G. Bayesian analyses were performed through the CIPRES Science Gateway the maximum time offered by that server (167 h) using the hybrid version (CIPRES Science Gateway v3.3) of MrBayes 3.12 (Huelsenbeck & Ronquist, 2001; Ronquist & Huelsenbeck, 2003) with settings as in May-Collado & Agnarsson (2006) and Agnarsson & May-Collado (2008) with separate model estimation each gene and within protein coding genes for first, second, and third codon positions. The Markov chain Monte Carlo search for each matrix ran with four chains for approximately 18,000,000 sampling the Markov chain every 1,000 generations, and the sample points of the first 5,000,000 generations were discarded as ‘burnin’, after which the chains had reached approximate stationarity as determined by analysis in Tracer. Maximum likelihood analysess was done in Garli 2.0 (Zwickl, 2006) and MEGA6 on the focal matrix with same partitions as implemented in the Bayesian analysis. Due to computational constraints, GARLI was run for 100 replicates and MEGA6 for 100 bootstrap replications. Trees were first manipulated in via Mesquite 2.75v (Maddison & Maddison, 2011) and then rendered in Adobe Illustrator. Finally, all trees and matrices will be uploaded to Tree Base (submission 16862).

As the basal topology of our preferred tree differs from many recent studies, we performed a Shimodaira-Hasawa test (Shimodaira & Hasawa, 1999) to see if alternative topologies could be rejected. We tested topologies where (1) Didelphimorphia is sister to the remaining Metatheria, (2) where Ameridelphia is monophyletic, and (3) where Australian Australidelphia is monophyletic (that is, Microbiotheriidae is sister to other Australidelphia). The test was run in PAUP* (Swofford, 2003) under a GTR + I + G model, using one-tailed test (RELL bootstrap) and 1,000 bootstrap replicates, starting branch lengths were obtained using the Roger-Swofford approximation method for branch and branch length optimization using one dimensional Newton–Raphson.

Benchmark clades

An intuitive approach to evaluating if the species-level phylogeny will serve as a reliable comparative tool is the recovery of well supported clades from prior studies and undisputed taxonomic groups (May-Collado & Agnarsson, 2006; Agnarsson & May-Collado, 2008). This approach is valuable given the nature of this study. The matrix is based on available data from Genbank rather than markers chosen specifically for the question at hand; some may not be ideal to estimate ancient phylogenetic relationships, and many data are missing in the concatenated matrix. The recovery of benchmark clades is a simple ‘reality check,’ indicating that the analyses are not critically impeded by these shortcomings of the data. A total list of 22 benchmark clades is provided in Table 1.

Results

Benchmark clades

With the exception of Phalangeriformes, data partitions in general supported the majority of benchmark clades (Fig. 1). There were notable differences between the mtDNA partition and the remaining partitions. The mtDNA partition data alone resulted in a phylogenetic hypothesis in greater conflict with taxonomy and recent phylogenetic studies than did full concatenated partitions analyses and the nuDNA partition alone, particularly at lower taxonomic levels (Fig. 1 and Figs. S1–S5). In the analysis of the full concatenated matrix, many clades have low support and some species appear conspicuously misplaced, such as Marmosa tyleriana (Fig. S1). Excluding protamine from the nuclear partition in general resulted in the same relationships (Fig. S2). Removing from the concatenated analysis taxa with <10% character data, our focal analysis, in general, recovered the majority of benchmark clades as the full analysis, while support for many clades increased and no species are conspicuously misplaced (Figs 2–4). The Bayesian analysis with only protein coding genes shows the same relationships as described above. The placement of the Sminthopsis crassicaudata within Diprotodontia seems to be due to an error in classification; the sequence for IRBP was blasted in GenBank and was 100% similar to that of Petrogale xanthopus. However, this error did not impact the results of the overall results presented here. Finally, adding retroposon data to this analysis yielded nearly identical results.

Figure 1 Summary cladogram of all the analyses showing support for relationships among major clades within Metatheria.

Figure 2 Majority rule consensus of the Bayesian analyses using the focal concatenated character matrix (excluding species with less than 10% data coverage) for Ameridelphia.

Note that Micoureus was recently recognized as a subgenus of Marmosa (IUCN Red List of Threatened Species 2013.2). Node numbers are posterior probabilities, omitted when less than 50%. Extinct species are indicated with a cross. Photo credits: “Caenolestes condorensis” uploaded by Kennethgrima to http://mt.wikipedia.org/wiki/Stampa:Caenolestes_condorensis.jpg; “Opposum 2” by Cody Pope (retrieved from http://commons.wikimedia.org/wiki/File:Opossum_2.jpg under a CC BY SA 2.5 license).

Figure 3 Majority rule consensus of the Bayesian analyses using the focal concatenated character matrix for Australasian marsupials: Notoryctemorphia, Peramelemorphia, and Dasyuromorphia.

Note that the species Antechinus naso, Antechinus melanurus and Paramurexia rothschildi are now recognized as members of the genus Murexia (IUCN Red List of Threatened Species 2013.2). Node numbers are posterior probabilities, omitted when less than 50%. Photo credits: “Notoryctes typhlops” by Rosa Catherine Fiveash (retrieved from http://en.wikipedia.org/wiki/Southern_marsupial_mole#mediaviewer/File:Notoryctes_typhlops.jpg); “Perameles gunni” uploaded by JJ Harrison (retrieved from http://en.wikipedia.org/wiki/Eastern_barred_bandicoot#mediaviewer/File:Perameles_gunni.jpg under a CC BY-SA 3.0 license); “Sarcophilus harrisii taranna” uploaded by JJ Harrison (retrieved from http://en.wikipedia.org/wiki/Tasmanian_devil#mediaviewer/File:Sarcophilus_harrisii_taranna.jpg under a CC BY-SA 3.0 license); “Myrmecobius fasciatus” uploaded by Martin Pot (retrieved from http://commons.wikimedia.org/wiki/Myrmecobius_fasciatus#mediaviewer/File:Numbat.jpg under a CC BY-SA 3.0 license).

Figure 4 Majority rule consensus of the Bayesian analyses using the focal concatenated character matrix for Australasian marsupials: Diprotodontia.

Node numbers are posterior probabilities, omitted when less than 50%. Photo credits: “Monito del Monte ps6” uploaded by José Luis Bartheld (retrieved from http://en.wikipedia.org/wiki/Monito_del_monte#mediaviewer/File:Monito_del_Monte_ps6.jpg under a CC BY 2.0 license); “Koala climbing tree” uploaded by Diliff (retrieved from http://en.wikipedia.org/wiki/Koala#mediaviewer/File:Koala_climbing_tree.jpg under a CC BY-SA 3.0 license); “Trichosurus vulpecula 1” uploaded by JJ Harrison (retrieved from http://en.wikipedia.org/wiki/File:Trichosurus_vulpecula_1.jpg under a CC-BY-SA-2.5 license); “Kangaroo and joey03” uploaded by Fir0002 (retrieved from http://it.wikipedia.org/wiki/Macropus_giganteus#mediaviewer/File:Kangaroo_and_joey03.jpg under a GNU Free Documentation License 1.2); “Sugies03 hp” uploaded by Anke Meyring (retrieved from http://en.wikipedia.org/wiki/Petaurus#mediaviewer/File:Sugies03_hp.jpg under a CC-BY-SA-2.5 license).

Phylogenetic relationships among orders

The deeper level phylogenetic relationships among orders were consistent between analyses (Fig. 1). These analyses did not support the monophyly of Ameridelphia, but instead placed Paucituberculata as the lineage sister to the remaining marsupials, composed of Didelphimorphia plus Australidelphia. In addition, our results did not support the clade Eometatheria as defined by Kirsch, Lapointe & Springer (1997), a clade consisting of Microbiotherida and all autralidelphians excluding peramelians, or as defined by Mckenna & Bell (1997), a clade consisting of the Dasyuromorphia, Notorycterimorphia, Peramelemorphia, Diprotodontia but excluding Microbiotherida. The South American Microbiotheria was the sister lineage of Diprotodontia in all analyses, though this relationship was never strongly supported. The relationship between Dasyuromorphia and Peramelemorphia to the exclusion of Notorycterimorphia was moderately supported by some of the analyses (Fig. 1). These results do not support a sister group relationship between the Peramelemorphia and the Diprotodontia expected of the two members of the Grandorder Syndactyli proposed by Mckenna & Bell (1997).

Phylogenetic relationships within orders

Paucituberculata

Within Paucituberculata, the genus Caenolestes was sister to a clade containing Lestoros and Rhyncholestes was strongly supported in all concatenated and mtDNA analyses (Figs. S1 and S4).

Didelphimorphia

Within Didelphimorphia, the monophyly of subfamily Caluromyinae as recognized by Voss & Jansa (2009) containing Caluromys and Caluromysiops was supported (Figs. S1–S3). The only extant species of the subfamily Hyladelphinae (Voss & Jansa, 2009), the Kalinowski’s mouse opossum (Hyladelphys kalinowskii), was sister to Didelphinae, and the only member of the subfamily Glironiinae, the bushy-tailed opossum; Glironia venusta was placed in most analyses basal to the clade containing the Hyladelphinae and the Didelphinae (Fig. 2 and Figs. S1–S3). The monophyly of the subfamily Didelphinae (Marmosa, Monodelphis, Tlacuatzin, Metachirus, Chironectes, Didelphis, Lutreolina, Philander and Chacodelphys, Cryptonanus, Gracilinanus, Lestodelphys, Marmosops, and Thylamys) was supported by the all analyses except the mtDNA partition (Fig. S4). The monophyly of the genera Thylamys, Cryptonatus, Marmosops, Monodelphis, Gracilinanus, Philander and Marmosa (plus subgenus Micoureus) were supported by most analyses (Fig. 2 and Figs. S1–S6). Other interesting relationships include the sister relationship of the genera Cryptonanus and Gracilinanus, the placement of the grayish mouse opossum (Tlacuatzin canescens) sister to Marmosa, and the close relationship of the Patagonian opossum (Lestodelphys halli) with members of the genus Thylamys, and the basal position of the Virginia opossum (Didelphis virginiana) to Philander and the other taxa of Didelphis (Fig. 2 and Figs. S1–S6).

Peramelemorphia

Within Peramelemorphia, the monophyly of the family Peramelidae was corroborated in three analyses (Fig. 1). The extinct member of the family Chaeropodidae the pig-footed bandicoot (Chaeropus ecaudatus) was placed as sister to the remaining peramelemorphians in the full multigene and mtDNA analyses with strong support (Figs. S1 and S4). In contrasts, it was placed in the nuDNA partition within the Peramelidae either sister to Echymiperinae (Fig. 3) or within the Microperoryctes (Figs. S2–S3) or within Diprotodontia (Fig. S6) but the support for these placements was low. In six of the analyses the greater bilby, Macrotis lagotis (Thylacomyidae) was placed as sister to the remaining peramelemorphians (Fig. 3 and Figs. S1, S4, S6). The monophyly of the genera Echymipera, Peroryctes, Isoodon and Perameles was supported by all analyses. The concatenated analyses differed from the mtDNA partition in the relations among genera. The most supported relationship was ((Echymipera + Microperoryctes) Peroryctes)+(Isoodon + Perameles) (Fig. 3 and Figs. S1 and S6).

Dasyuromorphia

Within Dasyuromorphia, the monophyly of the families Dasyuridae, Myrmecobiidae, and Thylacinidae was supported by most analyses with the interrelationships (Thylacinidae (Myermecobiidae, Dasyuridae)) (Fig. 1). In two analyses, the extinct Tasmanian wolf (Thylacinus cynocephalus) was placed within the subfamily Dasyurinae, but this relationship was poorly supported (Figs. S2 and S3). The monophyly of the two Dasyuridae subfamilies Dasyurinae and Sminthopsinae was also supported by most analyses, as well as the two Dasyurinae tribes Dasyurini and Phascogalini and the two Sminthopsinae tribes Planigalini and Sminthopsini (Fig. 3 and Figs. S1–S4). In most of the analyses, the phylogenetic position of Ningaui and the Kultarr (Antechinomys laniger) rendered the genus Sminthopsis paraphyletic (Fig. 3 and Figs. S1–S4). Also, highly consistent across analyses is the phylogenetic position of the Mulgara (Dasycercus cristicauda) as sister to the Ningbing fals antechinus, Pseudantechinus ningbing also rendering the later paraphyletic (Fig. 3 and Figs. S1–S4). The monophyly of the genera Planigale, Murexia, Antechinus, Phascogale, Myoictis, and Dasyurus was supported by most analyses (Fig. 3 and Figs. S1–S4).

Diprotodontia

With the exception of Phalangeriformes, the monophyly of other suborders, Vombatiformes (Vombatidae + Phascolarctidae) and Macropodiformes (Hypsiprymnodontidae + Macropodidae + Potoroidea) were supported by all analyses. The monophyly of superfamily Macropodidea was also supported (Fig. 1). The position of Vombatiformes as sister to the remaining Diprododontia was supported by all analyses except the mtDNA partition (Fig. 4 and Figs. S1–S4). Within the superfamily Petauroidea (Acrobatidae, Tarsipedidae, Pseudocheiriidae, and Petauridae), all partitions except mtDNA supported the position of the family Acrobatidae as the basal petauroids (Fig. 4 and Figs. S1–S3). Tarsipedidae was sister to a clade containing Petauridae and Pseudocheiridae. Within Petauridae, the placement of the Gymnobelideus sister to the genus Dactylopsila to the exclusion of Petaurus was strongly supported by both concatenated and the mtDNA partitions (Fig. 4 and Figs. S1–S4). The monophyly of the Pseudocheiridae was strongly supported, as well as that of the subfamilies Pseudocheirinae (Pseudochirulus, Pseudocheirus), Hemibelidinae (Hemibelideus, Petauroides) and Pseudochiropsinae (Petropseudes, Pseudochirops). The monophyly of genus Pseudochirops was not supported is most analyses (Fig. 4 and Figs. S1–S4). The superfamily Petauroidea is the sister group of Macropodiformes, and Phalangeroidea is the sister group to the clade containing Phalangeridae and Burramyidae (Fig. 1). Within the Phalangeridae, the Sulawesi cuscus (Strigocuscus celebensis) was the sister species of the Sulawesi bear (Ailurops ursinus) in all partitions, rendering the genus Strigocuscus paraphyletic, and the scaly tailed opossum Wyulda squamicaudata was sister to Trichosurus (Fig. 4 and Figs. S1–S4). The three kangoroo families Potoroidea, Macropodidae, and Hypsiprymnodontidae were monophyletic and their interrelationships Hypsiprymnodontidae (Potoroidea (Macropodidae) were also supported by all partitions except mtDNA (Fig. 4 and Figs. S1–S4). Within Potoroidea, the Rofous bettong, Aepyprymnus rufescens was sister to Bettongia to the exclusion of Potorous (Fig. 4 and Figs. S1–S3). Within Macropodidae, the monophyly of Macropus and Petrogale, was not supported by most analyses (Fig. 4 and Figs. S1–S3). In the full multigene analysis, Bennett’s tree kangaroo, Dendrolagus bennettianus, was the basal species in macropodids (Fig. S1), but for the pruned multigene analysis and nuclear analyses it was instead sister to all Phalanger species (Fig. 4 and Figs. S2 and S3).

Discussion

Our phylogenetic analyses include the largest taxon sampling of extant marsupial species to date, accounting for approximately 80% of the marsupial diversity currently recognized in Mammal Species of the World (Wilson & Reeder, 2005) or by the 2012 IUCN Red List of Threatened Species (http://wwwiucnredlistorg/). Overall, analyses resulted in phylogenetic trees congruent among partitions and in agreement with current phylogenetic knowledge, other than the basal arrangement of the American orders (see below). The mtDNA partitions resulted in deeper level relationships that are inconsistent with existing knowledge, thus the mtDNA data seems to be partially misleading, especially regarding deeper level relationships (Nilsson et al., 2010). However, the mtDNA data corroborates most of the shallower clades, and nuclear data signal appears to trump the mtDNA data in the combined analysis, resulting in hypotheses mostly consistent with taxonomy and current phylogenetic understanding. Missing data and ambiguity in the protamine data does not seem to have a strong impact on the analysis, but these factors in part explain low support for various clades in the full dataset analysis. Support in our focal analysis, excluding species with most missing data, was higher for many clades (Figs. 2–4). This finding is not surprising; it is well documented that placement of taxa with excessive (>90%) missing data can be problematic.

Root of the marsupial tree

As in most recent phylogenetic analyses, our result fails to support a monophyletic Ameridelphia, a clade only supported by mtDNA in this study and a few other studies (Retief et al., 1995; Springer, Westerman & Kirsch, 1994; Springer et al., 1998). All analyses resulted in the same basal arrangement, (Paucituberculata (Didelphimorphia, Australidelphia)). This conflicts with prior analyses in that the placement of Paucituberculata and Didelphimorphia are switched (Cardillo et al., 2004; Nilsson et al., 2010). However, a likelihood ratio test demonstrated that the alternative topologies consistent with recent studies are not significantly ‘worse’ explanations of the sequence data. For example, topologies consistent with the retroposon study of Nilsson et al. (2010) and topologies with monophyletic Ameridelphia have only slightly lower likelihoods than the preferred tree (see Table 2). Amrine-Madsen et al. (2003) also could not discriminate between these hypotheses, but favored rooting at the base of Didelphimorphia due to the long-branch leading to caenolestids. The sister relationship of Paucituberculata to Australasian marsupials is also supported by morphological characters (Horovitz & Sánchez-Villagra, 2003) and molecular studies (Baker et al., 2004; Kirsch, 1977; Kirsch, Lapointe & Springer, 1997).

Phylogenetic relationships among orders

Our results further place Microbiotheriidae within Australidelphia, sister to Diprotodontia, and this relationship is supported independently by the mtDNA and nuDNA. This result is congruent with a previous morphological study that suggested Dromiciops and diprotodontians shared a common ancestor that was hypothesized to have had a prehensile tail and an opposable hallux (Horovitz & Sánchez-Villagra, 2003). These taxa also share sperm morphology (Gallardo & Patterson, 1987) and have been supported previously using sequence data (Kirsch et al., 1991), and in a supertrees analysis (Cardillo et al., 2004). However, this arrangement is in conflict with some recent molecular studies (Asher, Horovitz & Sánchez-Villagra, 2004; Beck, 2008; Nilsson et al., 2010). In particular, the recent retroposon analysis of Nilsson et al. (2010) Microbiotheriidae is strongly supported as sister to the Australasian Australidelphia. The latter hypothesis is more congruent with the geographical distribution, placing all Australasian species in a clade and suggesting a single origin of Australasian marsupials. The source of this conflict between sequence and retroposon data is unclear. Future work should profitably focus on adding single-copy nuclear markers, retroposon data for a larger taxon sample, and ultimately employing rich sources of data through next generation sequencing techniques to solidify Marsupialia phylogenetics.

We find a moderately supported relationship between the orders Peramelemorphia and Dasyuromorphia to the exclusion of Notorycterimorphia, similar to previous studies (Baverstock, Kri & Birrell, 1990; Kullander, Carlson & Hallbook, 1997; Springer, Kirsch & Case, 1997; Beck, 2008; Colgan, 1999; Nilsson et al., 2010). The placement of Notoryctes has been controversial, with several studies supporting the genus as sister to Dasyuromorphia (Amrine-Madsen et al., 2003; Burk et al., 1999; Springer et al., 1998) while others as sister to Peramelemorphia (Baker et al., 2004). Finally, we did not find support for the Eometatheria as contra a prior study combining molecular and morphological data (Asher, Horovitz & Sánchez-Villagra, 2004).

Phylogenetic relationships within orders

Paucituberculata and Didelphimorphia

As found by recent molecular and morphological studies, the two Ameridelphian orders Paucituberculata and Didelphimorphia form a grade rather than a clade (Horovitz & Sánchez-Villagra, 2003; Nilsson et al., 2010). However, we cannot reject the monophyly of Ameridelphia. Our results corroborate Voss & Jansa’s (2009) combined morphological and molecular study where Glirona is placed in a monotypic subfamiliy Glironiinae, whereas Caluromys and Caluromysiops remained in the subfamily Caluromyninae. In addition, Hyladelphys was not grouped with other didelphids, thus we support its placement within a subfamily of its own Hyaldephinae (Voss & Jansa, 2009). However, not all analyses supported Didelphinae tribes (Marmosini, Metachirini, Didelphini, and Thylamyini). The species of the genus Cryptonanus (Voss, Lunde & Jansa, 2005) were formerly included in the genus Gracilinanus by Gardner & Creighton (1989). Previous studies have found it difficult to establish the phylogenectic position of Cryptonanus (Jansa, Forsman & Voss, 2006; Flores, 2009); however, with the exception of the mtDNA analyses, we found strong support for a sister relationship between Cryptonanus and Gracilinanus. A close phylogenetic position of the greyish mouse opossum, Tlacuatzin canescens (formely Marmosa canescens) with species of the genus Marmosa (and subgenus Micoureus) is also supported, as has been found in previous studies combining molecules and morphology (Flores, 2009; Voss & Jansa, 2009). The basal placement of the Peruvian gracile mouse opossum, Hyladelphys kalinowskii in the Didelphinae, is also supported by morphological and molecular analyses (Flores, 2009), as well as the close phylogenetic relationship between Patagonian opossum Lestodelphys halli and the genus Thylamys (Cardillo et al., 2004; Jansa, Forsman & Voss, 2006; Flores, 2009).

Dasyuromorphia

Our results for Dasyuromorphia are largely congruent with previous studies in supporting the monophyly of dasyuromorphian families and subfamilies. The phylogenetic position of the Numbat (Myrmecobius fasciatus) and the Tasmanian wolf (Thylacinus cynocephalus) are traditionally controversial. In this study, most analyses strongly supported the basal placement of the Tasmanian wolf and the Numbat to all dasyuromorphians. This phylogenetic arrangement has been previously supported by morphological (Wroe et al., 2000) and combined analyses (Beck, 2008) including a supertree analysis (Cardillo et al., 2004). The position of the Ningaui, and the Kultarr (Antechinomys laniger) within Sminthopsis agrees with previous studies (Archer, 1975; Blacket et al., 1999; Cardillo et al., 2004) and suggests that taxonomic changes are necessary to accommodate this phylogenetic perspective, In the supertree analysis by Cardillo et al. (2004), the Kultarr was referred as Sminthopsis laniger and was placed as basal to a clade containing Ningaui and Sminthopsis species. Also highly consistent across analyses in this study is the phylogenetic position of the Mulgara, Dasycercus cristicauda as sister to the Ningbing false antechinus, Pseudantechinus ningbing, contrasting the results by Cardillo et al. (2004).

Peramelemorphia

The results from the complete multigene and mtDNA data sets are congruent with recent studies placing the extinct pig-footed bandicoot (Chaeropus ecaudatus) sister to other bandicoots (Cardillo et al., 2004; Westerman et al., 2012). Our studies also support a proposed early divergence of the greater bilby, Macrotis lagotis, from other bandicoots (Cardillo et al., 2004; Meredith, Westerman & Springer, 2008a; Meredith, Westerman & Springer, 2008b; Westerman, Meredith & Springer, 2010). The greater bilby is known to be drastically different in their genetic make-up, in that it possesses sixteen autosomes and a multiple sex-chromosome system (Westerman, Meredith & Springer, 2010), contrasting with the typical bandicoot chromosome set of 2n = 14 (Westerman et al., 2012). In addition, our results support the close phylogenetic relationships between the two New Guinean subfamilies Echymiperinae and Peroryctinae (Meredith, Westerman & Springer, 2008a; Meredith, Westerman & Springer, 2008b; Westerman et al., 2012).

Diprotodontia

Like previous studies, our analysis strongly supported the monophyly of Diprotodontia. With the exception of Phalangeriformes the monophyly of the suborders Vombatiformes and Macropodiformes was supported contrasting with one molecular study (Meredith, Westerman & Springer, 2009). Amrine-Madsen et al. (2003) found ambivalent support for the monophyly of Phalangeriformes, and other studies have been inconclusive (Springer & Kirsch, 1991; Kirsch, Lapointe & Springer, 1997; Burk et al., 1999). The basal position of Vombatiformes (Vombatidae + Phascolarctidae) is in agreement with previous molecular studies (Meredith, Westerman & Springer, 2009; Nilsson et al., 2010).

The monophyly of the superfamily Phalangeroidea and families Phalangeridae and Burramyidae was supported and in accordance with previous studies (Osborne, Christidis & Norman, 2002; Beck, 2008; Phillips & Pratt, 2008; Meredith, Westerman & Springer, 2009). Our results are also congruent with previous findings supporting the clade Petauroidea where the families Petauridae and Pseudocheiridae grouped together to the exclusion of Tarsipedidae and Acrobatidae (Osborne, Christidis & Norman, 2002; Kavanagh et al., 2004; Phillips & Pratt, 2008; Meredith, Westerman & Springer, 2009; Meredith et al., 2010). Within the family Petuaridae, our results contrast recent molecular studies and previous morphological studies, in that the genus Gymnobelideus grouped with to the genus Dactylopsila to the exclusion of Petaurus (Smith, 1984; Aplin & Archer, 1987; Springer, Westerman & Kirsch, 1994; Meredith et al., 2010). These results are more in accordance with the molecular analysis of Edwards & Westerman (1995). Like a previous study (Meredith et al., 2010), we find strong support for the monophyly of the family Pseudocheiridae and three subfamilies. Our results agree with Meredith et al. (2010) in that the genus Pseudochirops is paraphyletic due to the placement of the Australian rock-haunting ringtail possum Petropseudes dahli with New Guinean species of Pseudochirops. Our results are also in agreement with Meredith et al. (2010), in the genera placed in the clades corresponding to the Pseudocheirinae and Pseudochiropsinae in contrast to the assignments by Groves (2005).

Petauroidea grouped with Macropodiformes to the exclusion of Phalangeroidea, contrasting previous studies results (Phillips & Pratt, 2008; Meredith, Westerman & Springer, 2009). As shown by previous studies, the three kangaroo families are monophyletic (Osborne, Christidis & Norman, 2002; Meredith, Westerman & Springer, 2009; Phillips et al., 2013). One interesting result was the consistent placement of the Sulawesi cuscus, Strigocuscus celebensis, sister species to the Sulawesi bear, Ailurops ursinus, rendering the genus Strigocuscus paraphyletic. This result agrees with previous molecular studies (Meredith, Westerman & Springer, 2009). Within Macropodiformes, the family Hypsiprymnodontidae was sister to a clade consisting of Macropodidae and Potoroidae. This relationship agrees with a previous study (Burk, Westerman & Springer, 1998). Within Potoroidae the relationship between Bettongia and Aepyprymnus to the exclusion of Potorous was strongly supported similarly to previous support (Burk, Westerman & Springer, 1998). The monophyly of the genus Petrogale was strongly supported by this analysis but phylogenetic relationships among species contrast those proposed by Campeau-Péloquin et al. (2001).

Conclusions

Here, we offer a summary primary-data phylogeny for marsupials, including 80% of the known marsupial diversity, utilizing a Genbank data-mining approach. Overall results are consistent with previous phylogenetic studies and generally recover undisputed deeper level clades, suggesting the present phylogenetic hypotheses should serve as valuable tools for future taxonomic and comparative evolutionary studies.

Supplemental Information

Table S1 Species included in each analysis and their respective GenBank Accession number.

Note that Micoureus was recently recognized as a subgenus of Marmosa, and Antechinus naso, Antechinus melanurus, and Paramurexia rothschildi are now recognized as members of the genus Murexia (IUCN Red List of Threatened Species 2013.2).

Click here for additional data file.

Table S2 Detailed results from the Shimodaira-Hasagawa test evaluating alternative topologies compared to the focal analysis (tree1 = focal analysis, tree2 = Ameridelphia monophyletic, tree 3 = Didelphimorphia basal, tree4 = Retroposon tree by Nilsson et al., 2010)

Click here for additional data file.

Figure S1 Metatheria majority rule consensus of the Bayesian analyses using the full concatenated character matrix

Note that Micoureus was recently recognized as a subgenus of Marmosa, and Antechinus naso, Antechinus melanurus, and Paramurexia rothschildi are now recognized as members of the genus Murexia (IUCN Red List of Threatened Species 2013.2).

Click here for additional data file.

Figure S2 Metatheria majority rule consensus of the Bayesian analyses using the full concatenated character matrix excluding protamine

Note that Micoureus was recently recognized as a subgenus of Marmosa, and Antechinus naso, Antechinus melanurus, and Paramurexia rothschildi are now recognized as members of the genus Murexia (IUCN Red List of Threatened Species 2013.2).

Click here for additional data file.

Figure S3 Metatheria majority rule consensus of the Bayesian analyses using the nuDNA concatenated character matrix excluding protamine

Note that Micoureus was recently recognized as a subgenus of Marmosa, and Antechinus naso, Antechinus melanurus, and Paramurexia rothschildi are now recognized as members of the genus Murexia (IUCN Red List of Threatened Species 2013.2).

Click here for additional data file.

Figure S4 Metatheria majority rule consensus of the Bayesian analyses using the mtDNA concatenated character matrix

Note that Micoureus was recently recognized as a subgenus of Marmosa, and Antechinus naso, Antechinus melanurus, and Paramurexia rothschildi are now recognized as members of the genus Murexia (IUCN Red List of Threatened Species 2013.2).

Click here for additional data file.

Figure S5 Metatheria majority rule consensus of the Bayesian analyses using a matrix excluding non-protein coding genes (12S and 16S) plus the ambiguously aligned protamine

Note that Micoureus was recently recognized as a subgenus of Marmosa, and Antechinus naso, Antechinus melanurus, and Paramurexia rothschildi are now recognized as members of the genus Murexia (IUCN Red List of Threatened Species 2013.2)

Click here for additional data file.

Figure S6 Maximum likelihood tree using excluding species with less than 10% data coverage

Note that Micoureus was recently recognized as a subgenus of Marmosa, and Antechinus naso, Antechinus melanurus, and Paramurexia rothschildi are now recognized as members of the genus Murexia (IUCN Red List of Threatened Species 2013.2).

Click here for additional data file.

We thank our Academic Editor for PeerJ Xiaolei Huang, and reviewers Jonathan Fong and Deyan De for comments that greatly improved this manuscript.

Additional Information and Declarations

Competing Interests

Author Contributions

The authors declare there are no competing interests.

Laura J. May-Collado and Ingi Agnarsson conceived and designed the experiments, performed the experiments, analyzed the data, contributed reagents/materials/analysis tools, wrote the paper, prepared figures and/or tables, reviewed drafts of the paper.

C. William Kilpatrick conceived and designed the experiments, wrote the paper, reviewed drafts of the paper.

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
