# Peer review of "Mammals from ‘down under’: a multi-gene species-level phylogeny of marsupial mammals (Mammalia, Metatheria)"

_PeerJ, doi:10.7717/peerj.805_

## Round 0.1 · original submission · Major Revisions

While both reviewers think the manuscript is potentially publishable, they also provide valuable comments for improving the analyses and presentation of the results (incl. tables, figures, grammar ). I suggest the authors carefully address the questions and suggestions raised by the reviewers. I agree with Reviewer 1 that you should submit datasets related to your paper to appropriate public databases.

·

Basic reporting

The submission adheres to PeerJ policies.
Please check the grammar of your text, especially the use of punctuation. There are several places in the text where there are commas, periods, colons missing (I have pointed out some of these in the attached pdf file).

Experimental design

The experimental design is appropriate and the research question has been clearly stated (compiling and analyzing available data in GenBank to infer a marsupial phylogeny).
My suggestion here is to pick clear names for each of the datasets analyzed and maintain this name throughout the text and figures. For example the main dataset is called "full concatenated character matrix" (line 104), "all data combined" (line 292), or full concatenated character matrix (excluding species with less than 10% data coverage)" (lines 924-925).
Another suggestion would be to list all the datasets you tested, but emphasize which dataset your results focus on. You list ~8 different datasets, but most of your results and discussion focus on the "full concatenated character matrix (excluding species with less than 10% data coverage)".

Validity of the findings

These data are robust and the conclusions appropriate.

Additional comments

Here are some specific revisions that should be made (in addition to the ones stated above). Most are also annotated on the pdf.
1) (lines 71-85) This is an important paragraph showing why your study is new and important. Previously, the largest dataset was Cardillo et al (2004) with 260 taxa, and your study has 276. You have stated why your study is different, but you can emphasize this more. Say something about what a supertree approach is, and how your approach is better because you are reanalyzing data instead of combining results from other studies.
2) (lines 102-107) in addition to the number of taxa for each dataset, please include some information on how long the data matrix is (bp).
3) Is there a reason why you chose <10% data coverage as your cut-off? I assume there was a balance between increasing the coverage level and discarding taxa, but 10% seems low.
4) (line 122) how was stationarity checked?
5) (lines 130-282) benchmark clades: this section is very long and may not be needed as text. Including it as text makes it seem important to read, until you start reading and realize it is a list of references. I suggest trying to summarize this information in a table. For columns, try "clade", "description", "references". Or you could try to make groups of data types for columns.
6) Please submit your datasets to TreeBase
7) in the methods, you talk about how you added the retroposon data to the full and 251-taxon matrix, but there is no discussion of the results. Were these results different from the datasets without retroposon data?
8) (lines 916-918) Table 1 caption. Currently the caption describes the results, not what the table is. Suggestion in pdf.
9) Table 1: don't fill in all the boarders for the table--makes it hard to read. Instead, use 3 horizontal lines (look at other papers for an example). Also, align the numbers, and don't double-space the text
10) Figure 1: For the squares showing support values, be consistent in the formatting. Will you put white lines between the squares or not? Also, find some way to denote the clades you are using as benchmark clades, such as putting the group names in bold.
11) Table S1: standardize the font and font size.

Reviewer 2 ·

Basic reporting

Dear editor,
Thank you very much for inviting me to review this manuscript. This study presented a detailed investigation on the species-level phylogeny of marsupial mammals (Mammalia, Metatheria). It is the first Methaterian species-level phylogeny, which included 80% of the extant marsupial species. They aim to provide a summary phylogeny that will serve as a tool for comparative research. Although there is no new data produced in the present study, authors compiled the largest dataset to date, which could provide as a comprehensive phylogenetic framework for future studies.

This manuscript is potentially suitable for Peer J, while the following aspects should be considered before it is acceptable for publication.

I hope that my comments can help the authors to improve this paper. My review is mostly focusing on general issues and methods, as I am not an expert on marsupial species.

Sincerely yours: Deyan De

Experimental design

Since there are no new data produced in the present study, I strongly suggest the authors to strengthen their methods for phylogenetic reconstruction as well as testing the sensitivity of alternative phylogenetic topologies. The detailed information for the major references which produced the original data probably could be compiled in a table. Besides, several important references were used mistakenly, and in several place, they were totally absent. The figures are very beautiful, while information referred to their major clades is absent, which is difficult for readers.

Validity of the findings

It is the first Methaterian species-level phylogeny, which included 80% of the extant marsupial species. They aim to provide a summary phylogeny that will serve as a tool for comparative research.

Additional comments

Main text
1.Would it possible to illustrate the main changes of phylogenetic hypotheses in one figure? It will be more readable for people who are not very familiar with marsupial species and previous publications.
2.Detailed information for DNA fragments used in phylogenetic reconstruction is lacking in the main text, please give full spellings for their first appearance.
3.Topological structure of the tree is sensitive to alignment methods as well as the methods used for phylogenetic reconstruction, while in the present study, authors only chose one method to align these sequences. Bayesian inference was the single method used here to reconstruct phylogeny. I suggest the using of different methods to align sequences and reconstruct the tree.
4.Authors included several sequences of fossils in analysis, would it possible to illustrate these information on the tree?
5.Settings for Shimodaira–Hasegawa test were absent. Please give detailed information.
6.Please check the citation of Mafft.
7.How did you illustrate the trees? Please give detailed information on this.


Tables
1.Please give the topological structure of the phylogenetic hypotheses tested in ‘Shimodaira-Hasawa test’.
2.Please check the statistic results of Shimodaira–Hasegawa test, those values are not enough for interpreting this kind of test.
3.Please check the format of supplementary table 1.
4.The first column of the supplementary table 1 is mixed with names for orders, families and species, please separate them into independent columns.

Figures
1.Figure legends are too simple to interpret the content of figures.
2.All the supplementary trees are given without a scale, please check.
3.Annotations for major braches of the supplementary trees are absent.
4.Who photographed these animals? Number and Latin names for each figure in the right side of the tree are absent.
5.Posterior probabilities down to the second decimal point is a normal way for showing the robust of these nodes.
6.Posterior probabilities lower than 0.50 could be discarded.

References:
1.Please check the references strictly by following the format of Peer J.

---

## Round 0.2 · accepted · Accept

Thank the authors for addressing the questions from the reviewers, and submitting more supplementary data to make the ms more 'open'. I am happy to accept the manuscript now.